# FROM $f(x)$ AND $g(x)$ TO $f(g(x))$: LLMS LEARN NEW SKILLS IN RL BY COMPOSING OLD ONES

**Lifan Yuan**[1*], **Weize Chen**[2*], **Yuchen Zhang**[3,4], **Ganqu Cui**[3†], **Hanbin Wang**[4],
**Ziming You**[4], **Ning Ding**[2,3†], **Zhiyuan Liu**[2†], **Maosong Sun**[2], **Hao Peng**[1]
[1] University of Illinois Urbana-Champaign   [2] Tsinghua University
[3] Shanghai AI Laboratory   [4] Peking University
lifan4@illinois.edu   chenwz21@mails.tsinghua.edu.cn

## ABSTRACT

Does reinforcement learning (RL) teach large language models (LLMs) genuinely new skills, or does it merely activate existing ones? This question lies at the core of ongoing debates about the role of RL in LLM post-training. On one side, strong empirical results can be achieved with RL even without preceding supervised fine-tuning; on the other, critics argue that RL contributes little beyond reweighting existing reasoning strategies. This work provides concrete evidence that LLMs can acquire genuinely new skills during RL by composing existing ones, mirroring one of the central mechanisms by which humans acquire new cognitive skills (Anderson, 1982). To mitigate data contamination and other confounding factors, and to allow precise control over task complexity, we develop a synthetic framework for our investigation. Specifically, we define a skill as the ability to infer the output of a string transformation function $f(x)$ given $x$. When an LLM has already learned $f$ and $g$ prior to RL, our experiments reveal that RL enables it to learn unseen compositions of them $h(x) = g(f(x))$. Further, this compositional ability generalizes to more difficult problems such as compositions of $> 2$ functions unseen during RL training. Our experiments provide surprising evidence that this compositional ability, acquired on the source task, transfers to a different target task. This transfer occurs even though the model has never trained on any compositional problems in the target task, and the only requirement is that the model has acquired the target task's atomic skills before its RL training on the source. Our qualitative analysis shows that RL fundamentally changes the reasoning behaviors of the models. In contrast, none of the findings is observed in next-token prediction training with the same data. Our systematic experiments provide fresh insights into the learning behaviors of widely-used post-training approaches for LLMs. They suggest the value of building base models with the necessary basic skills.

## 1 INTRODUCTION

Reinforcement learning (RL) has achieved broad success in improving large language models (LLMs) on a variety of tasks especially reasoning (OpenAI, 2024; DeepMind, 2025), even directly building upon the base model without any preceding supervised fine-tuning (DeepSeek-AI et al., 2025). Despite the profound success, recent work finds the exploration of RL is impeded by the entropy collapse phenomenon (Cui et al., 2025b; Liu et al., 2025a; Yu et al., 2025), and the performance gaps between base and RL-trained models diminish as the number of samples ($k$) increases in pass@$k$ evaluations (Yue et al., 2025). In addition, some argue that the "aha moments" in RL training (OpenAI, 2024; DeepSeek-AI et al., 2025) are not emergent but merely the result of amplifying existing cognitive behaviors present in base models (Gandhi et al., 2025; Liu et al., 2025b; Zhao et al., 2025), which casts shadow on whether LLMs learn new skills during RL training (Wu et al., 2025a). Such observations diverge from established RL findings that predate LLMs, where models were trained from scratch and learned new skills (Silver et al., 2016; 2017; OpenAI et al.,

---

*Equal Contribution. Orders are determined randomly.   † Corresponding author.

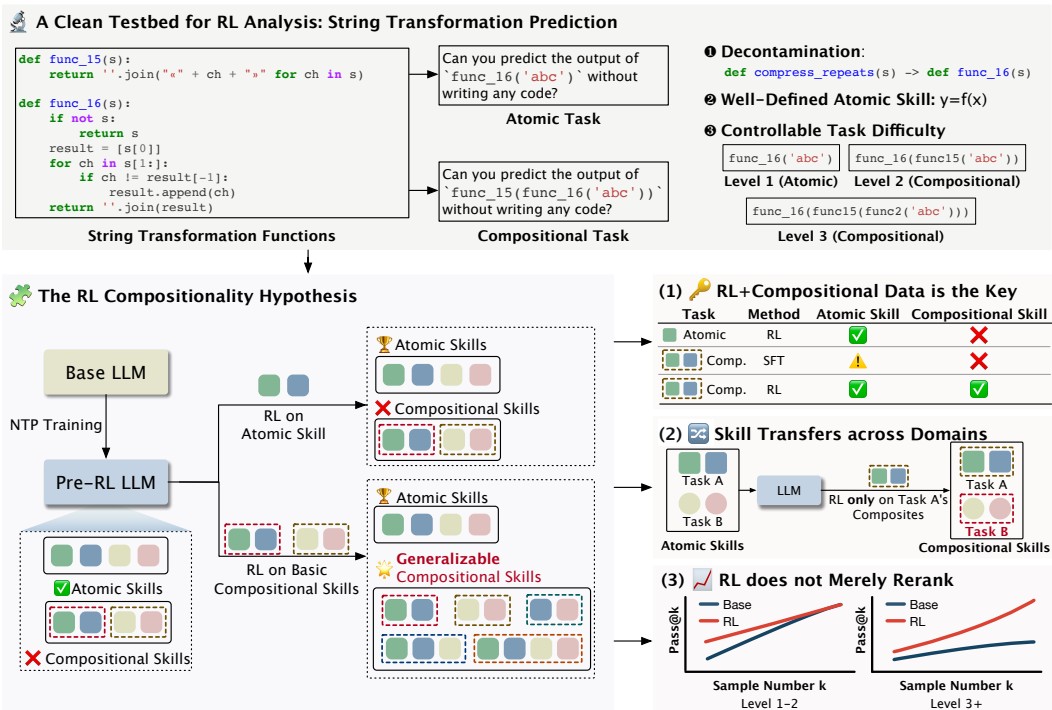

Figure 1: **An overview of our research framework and key findings. (Top)** We introduce a clean string transformation testbed to scientifically analyze RL's capabilities. **(Bottom-Left)** Our central RL Compositionality Hypothesis posits that training on simple composites with RL unlocks generalizable compositional skills. **(Bottom-Right)** Our experiments validate this, showing that: (1) compositional data combined with RL is the key ingredient for learning this new skill; (2) the learned skill transfers across domains; and (3) RL significantly improves difficult problems where the base model fails, while only reranking on problems it solves well.

2019). The fact that LLMs are pretrained on vast data prior to RL may contribute to these divergences and call for further investigation into the following important research questions: **(1) Does RL teach new skills to LLMs? (2) If so, how to incentivize it? (3) Are the skills generalizable?** Answering these questions will advance our understanding of LLM learning behaviors and inform the high-stakes trade-off in resource allocation between pretraining and post-training.

We provide concrete evidence that LLMs indeed learn new skills in RL, by composing and generalizing existing skills to solve more complex problems; For such learning to happen, there should be proper incentivization in RL. Our investigation is grounded in the cognitive skill acquisition process by humans, inspired by Anderson (1982), which argues that humans learn new skills by composing and then internalizing existing ones. Unlike prior works (Gandhi et al., 2025; Yue et al., 2025), we choose to construct a controlled synthetic framework that facilitates:

- **Decontaminated evaluation**: We design a string transformation prediction task with unique functions assigned meaningless identifiers (e.g., `func_16`) to prevent inference from function names. During RL, function definitions are hidden. The tasks will then be unsolvable without going through our atomic skills acquisition training. This setup enables us to investigate the RQs controlling for confounders.
- **Well-defined atomic and compositional skills**: We define atomic skills as single, non-decomposable transformations, and compositional skills as their nested combinations. For example, given input string $x$, `func_16(x)` represents an atomic skill, while `func_15(func_16(x))` requires compositional reasoning.
- **Controllable difficulty**: As each skill is instantiated as a Python function, we control difficulty of composition through the depth of nesting. As shown in Fig. 1, the model must perform deductive reasoning to give the output string after a given transformation, e.g., a Level-1 difficulty problem `func_16(x)` and a Level-2 one `func_16(func_15(x))`. Here the difficulty level is determined by the number of atomic functions composed.

With our framework and a two-stage training protocol that separates atomic from compositional skill acquisition, we conduct experiments with Llama-3.1-8B-Instruct (Dubey et al., 2024) and answer the RQs as follows:

- **RL teaches new compositional skills.** RL on Level-2 problems, receiving only correctness-based outcome rewards without reasoning demonstrations, substantially improves generalization on more difficult problems: performance on unseen Level-3 tasks improves from near-zero to 30%, and Level-4 to 15%. This generalization does not occur in a baseline trained with rejection fine-tuning (RFT) on the same Level-2 problems. This shows that RL enables the acquisition of compositional skills.
- **Both RL and compositional incentives are essential for skill acquisition.** In contrast to the substantial accuracy improvements from RL on Level-2 compositional problems, RFT on the same data and RL on Level-1 atomic problems both yield little improvements on problems higher than Level-2 (e.g., less than 1% improvement at Level-3). This may explain why Sun et al. (2025) conclude that RL does not promote compositional generalization, as their training includes no explicit incentive for composition.
- **The learning achieved by RL generalizes to held-out evaluation, more difficult problems, and even a different task.** All findings above are based on held-out evaluation of compositional problems consisting of atomic skills (functions) unseen in RL training. And as aforementioned, models RL-trained on Level 2 problems show non-trivial gains on problems up to Level 4. For cross-task transfer, compositional RL on the string task boosts accuracy on the unseen Level-3 Countdown problems to 35% for a model with the prerequisite Countdown atomic skills.

Our findings challenge the recent view that current RL with verifiable rewards (RLVR) (Lambert et al., 2025) merely utilizes reasoning patterns in base models rather than learning new reasoning abilities (Yue et al., 2025; Wu et al., 2025a). This view is based on the observation that the pass@$k$ performance gap between RL-trained and base models narrows as $k$ increases (Yue et al., 2025). We conjecture that this observation arises, at least in part, from evaluating and RL training on tasks where base models already achieve high pass@$k$, possibly due to pretraining on similar tasks that is beyond the control of most academic researchers; thus RL has little incentive to learn a skill that the base model already has. To confirm this conjecture, our experiments show that RL substantially improves pass@$k$ on challenging compositional problems where base model's pass@$k$ is near zero (See Fig. 5). This reveals what we term the "reranking illusion," namely aggregate metrics on mixed-difficulty benchmarks can mask genuine skill acquisition by conflating capabilities of different types. Our qualitative analysis confirms that RL fundamentally changes reasoning behavior. As shown in Fig. 6, compositional errors, i.e., ignoring composition and misunderstanding function relationships, drop substantially, while failures shift primarily to atomic prediction errors (55%). This behavioral transformation indicates genuine acquisition of compositional skills.

Our findings have important implications for LLM development and highlight RL's critical role in post-training, particularly its potential for easy-to-hard generalization and cross-task transfer. They call for closer coordination between base model development and post-training strategy from a skill acquisition perspective.

## 2 BACKGROUND

**The Recent Pessimistic View on Whether RL Teaches New Skills to LLMs.** RL in LLMs builds on a model pretrained on vast data. While supervised warm-starts are a common technique in traditional RL (Silver et al., 2016; Vinyals et al., 2019; De La Cruz Jr et al., 2019; Silva & Gombolay, 2021), the large-scale and general-purpose nature of LLM creates a different scenario. On one hand, this strong prior enables base LLM to sample reasonable rollouts and thus perform RL directly without any preceding supervised fine-tuning (DeepSeek-AI et al., 2025; Pan et al., 2025; Zeng et al., 2025); on the other hand, it becomes difficult to distinguish genuine skill acquisition from activation of existing capabilities during RL training.

Recent work tries to investigate this but uses loose definitions of "skill", often relying on proxies such as the continually increasing frequency of certain reasoning patterns (Gandhi et al., 2025; Zhao et al., 2025; Liu et al., 2025b) or the diminishing gaps between the pass@$k$ accuracy of models before and after RL, as shown in the bottom right chart in Fig. 1 (Yue et al., 2025; Liu et al., 2025a; Wu et al., 2025a; He et al., 2025; Wen et al., 2025; Zhu et al., 2025). Although these studies show

that RL activates behaviors already present in the base model, they did not directly prove that no new skill is learned during the process. Moreover, the pass@$k$ results can be misinterpreted for many reasons: (1) The causal relation between performance and each skill remains unclear, thus it is not guaranteed that everything learned can be translated into improvements in pass@$k$ accuracy on downstream tasks. (2) The evaluation tasks only provide an obscure overall view, lacking fine-grained analysis on problems of different difficulty levels or domains. (3) The result is confounded by the possibility that the model has limited room or incentive to learn new skills if it already performs well prior to RL, especially when RL is conducted on data that overlaps with or closely resembles the data used during next-token prediction (NTP) training. (Wu et al., 2025c; Shao et al., 2025; Wang et al., 2025; Cui et al., 2025a; Yu et al., 2025; Liu et al., 2025a; Wu et al., 2025b). Together, these highlight the urgent need for a deeper analysis of tasks through a clean framework, in which the skills are clearly defined and contribute to the performance causally, and evaluated in a finer granularity.

**Compositional Learning as a Testbed Grounded in Cognitive Skill Acquisition in Humans**. Although the pessimistic conclusions about RL in LLMs from prior works are debatable, they at least indicate that the success of RL depends on strong base models. This motivates our study of skill composition, where RL learns new abilities by leveraging those already acquired by the base model. Compositional reasoning provides an ideal framework for investigating skill acquisition because it naturally separates atomic knowledge, which mirrors how humans learn cognitive skills (Anderson, 1982). Notably, it is established in cognitive science that both composed skills and the meta-ability to learn composition are non-trivial new skills (Anderson, 1982; Lake et al., 2016). For clarity, we refer to learning new skills as the former throughout this paper. Learning compositional skills helps the model to generalize to more challenging problems and new domains beyond training data, which we will show later. In the field of AI, compositional reasoning has been widely studied before LLMs and has been considered a necessary property of generalization. (Fodor & Pylyshyn, 1988; Lake et al., 2016; Andreas et al., 2015). More recently, Yin et al. (2025) achieved compositional improvements through in-context learning rather than RL, while Sun et al. (2025) found that directly RL in atomic skills fails in compositional generalization. Comparing the two works, we conjecture that an explicit incentive to composition is necessary.

## 3 RESEARCH FRAMEWORK

In this work, we define "new skills" as novel reasoning strategies that enable models to solve previously unsolvable problems through systematic combination of existing capabilities. We address three critical research questions: **(1) Does RL teach new skills to LLMs? (2) If so, how to incentivize it? (3) Are the learned skills generalizable?**

**Hypothesis 1** (The RL Compositionality Hypothesis). *Once a model has acquired the necessary atomic, non-decomposable skills for a task through NTP training, RL with proper incentivization can teach the model to learn new skills by composing atomic skills into more complex capabilities.*

### 3.1 TASK DESIGN: DEDUCTIVE REASONING ON STRING TRANSFORMATION PREDICTION

To test our hypothesis while avoiding confounders from data contamination and unclear skill boundaries, we design a controlled synthetic task with the following properties: (1) Atomic skills are well defined so that models can learn the fundamental skills separately before RL. Each string transformation function has clear, deterministic behaviors that can be learned independently. (2) Task difficulty can be controlled by adjusting the compositional complexity of the atomic skills, allowing us to test generalization across complexity levels. (3) RL and evaluation tasks do not appear in the LLM pretraining corpus, ensuring that improvements stem from learning rather than memorization.

**Task Definition**. Specifically, our task involves deductive reasoning on string transformations. Given an input string $x$ and a composition of deterministic transformation functions such as $f(\cdot)$ and $g(\cdot)$, models must predict the output string after applying the specified transformation (e.g., $y = f(g(x))$). We construct 25 unique string transformation functions as atomic skill spanning various computational patterns including character manipulation, reordering, filtering, and structural modifications (see Appendix §D for complete specifications). To mitigate potential contamination, we assign meaningless identifiers to string functions as shown in Fig. 1, so that it is impossible to infer the functionality with function names only.

**Difficulty Level**. We control compositional complexity through **Difficulty Levels** corresponding to nesting depth, with Level $n$ involving $n$-function composition. For instance, Level 1 involves single function application (e.g., `func_16(x)` as shown in Fig. 1), while Level 2 involves two-function composition (e.g., `func_16(func_15(x))`). The controlled difficulty provides a fine-grained inspection of model performance, rather than a vague overall number as adopted in prior work (Yue et al., 2025; Liu et al., 2025a; Wu et al., 2025a).

## 3.2 TRAINING AND EVALUATION PROTOCOL

Training consists of two stages to separate atomic skill acquisition from compositional skill learning, simulating realistic post-training pipelines.

**Stage 1 Training: Atomic Skills Acquisition via RFT**. Models learn "atomic skills" in this stage via rejection fine-tuning Dong et al. (2023). Specifically, we collect training data by prompting the model with explicit function definitions to generate correct reasoning trajectories. However, we remove these definitions from the prompts during the fine-tuning process. This compels the model to predict the output based solely on the function identifier, ensuring it internalizes the function's behavior, defined here as the atomic skill, rather than relying on in-context instructions. Crucially, the data collection phase of this stage is the only time models are exposed to the function implementations. An example can be found in Fig. 7.

**Stage 2 Training: Compositional Skill Training via Either RFT or RL**. In this stage, models see only function names and compositions, such as `func_2(func_16(x))`, with function definitions hidden. See Fig. 8 for examples. This forces reliance on internalized atomic knowledge while learning systematic composition. We compare two approaches: (1) **Composition via online RL** provides models with binary rewards based on output correctness and updates through Group Relative Preference Optimization (GRPO) Shao et al. (2024), testing whether RL is necessary for the acquisition of compositional skills. (2) **Composition via offline RFT** trains models with NTP on correct reasoning trajectories for compositional problems, serving as a baseline to examine whether exposure to compositional examples alone enables composition.

We use Llama-3.1-8B-Instruct, which is identified as a cleaner testbed for RL by recent work (Shao et al., 2025; Agarwal et al., 2025; Wu et al., 2025b), to further minimize the effect of data contamination besides our string tasks. For more details, please refer to Appendix A.

**Held-out, Easy-to-Hard, and Cross-Task Evaluation**. We assess generalization using rigorous held-out evaluation. In Stage 1, models are trained on all 25 atomic functions (Appendix D). In Stage 2, the functions are partitioned into two disjoint sets: the model trains only on compositions from one set, while the other is held out for evaluation. We test model generalization across various difficulty levels, using Countdown (Gandhi et al., 2024; Pan et al., 2025) as a testbed for task transfer.

## 4 RL AS A PATHWAY TO GENERALIZABLE SKILL ACQUISITION

### 4.1 LLMs ACQUIRE NEW COMPOSITIONAL SKILLS DURING RL

Our first experiment directly test our RL Compositionality Hypothesis (Hypothesis 1). To do so, we start from an identical Stage 1 base model and apply three different Stage 2 training configurations, allowing us to isolate the impact of incentivizing composition during RL: (1) **RL Level 1**, trained only on atomic tasks; (2) **RL Level 2**, trained only on two-level compositions; and (3) **RL Level 1+2**, trained on a uniform mix. We then evaluate their ability to generalize to held-out tasks from Level 1 up to Level 6, testing whether they can solve problems with unseen function compositions and higher nesting levels than seen in RL training.

As shown in blue curves in Fig. 2, training on Level 1 alone leads to high accuracy on Level 1, peaking at around 90%, but fails to generalize. Its accuracy on Level 2 task remains below 25%, and on Level 3 through 6, it is consistently near *zero*. This demonstrates that learning only the atomic skills through RL is insufficient for learning effective composition.

In contrast, incorporating compositional tasks into RL training yields transformative results. Both the RL Level 2 and RL Level 1+2 models demonstrate strong performance to generalize to problems with nesting depths exceeding their training data. On Level 3, their accuracy improves from

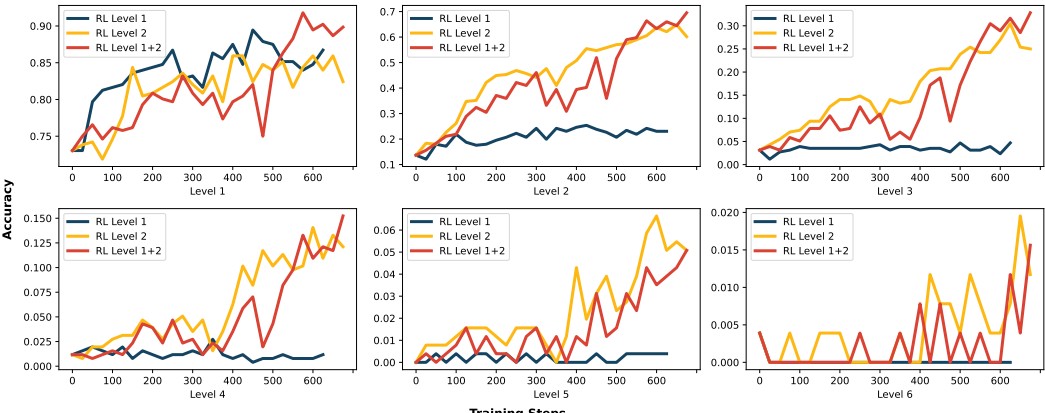

Figure 2: **Test Accuracy on held-out tasks vs. RL training steps, each related to one held-out task difficulty level.** The dark blue curve indicates that training on atomic skills alone (RL Level 1) yields nearly no compositional ability on held-out functions. In contrast, including Level 2 data in RL unlocks strong generalization to more complex problems (Levels 3-6).

5% to around 30%, and from 1% to 15% on Level 4, which are all significant improvements over the RL Level 1 model. And this trend continues on even Level 5, indicating both models learn a generalizable principle of compositional reasoning rather than merely memorizing solutions. This validates our hypothesis that RL can teach genuinely new skills, but only when the training objective explicitly incentivizes their use. These results provide us with evidence to answer RQ1:

> **TAKEAWAY 1**
>
> RL on compositional data teaches new skills that generalize to unseen compositions of known atomic skills.

## 4.2 RL is the Key Ingredient to the New Compositional Skills

Our previous experiment shows that compositional data is necessary for RL to teach new compositional skills, but **can a supervised method, such as RFT, achieve the same results as RL when given the exact same compositional (Level 2) data?** To address this question, we train a model with iterative RFT on the same Level 2 problems and conduct a head-to-head comparison against the RL Level 2 model from §4.1, with both having started from the identical Stage 1 base model.

The results in Fig. 3 show a significant difference in performance from Fig. 2. The RFT model's accuracy is significantly worse than RL across all compositional levels and has only marginal improvement over the first iteration. For example, on Level 3 it never surpasses 2.6%. In contrast, the RL Level 2 model achieves 64% on Level 2 and 27% on Level 3, significantly outperforming the RFT model. Surprisingly, the RFT model attains only 15% accuracy on Level-2 problems. This indicates that RFT fails to generalize even to held-out compositional problems of the same difficulty as its training data, let alone higher difficulties. These results provide the evidence to answer RQ2:

> **TAKEAWAY 2**
>
> RFT, even with on compositional data, is suboptimal for learning compositional skills; RL, in addition to compositional training data, is another important factor in learning generalizable compositional skills.

## 4.3 Compositional Skills Learned in RL are Transferable, but Atomic Skills are Prerequisites

While our experiments demonstrate that RL can teach generalizable compositional skills *within* a task, collecting compositional RL data for every new domain is impractical. We therefore test the transferability of the learned compositional skill. Specifically, we conjecture that RL enables models to compose atomic skills on Task B after learning composition on Task A, if the model has already acquired the necessary atomic skills for Task B.

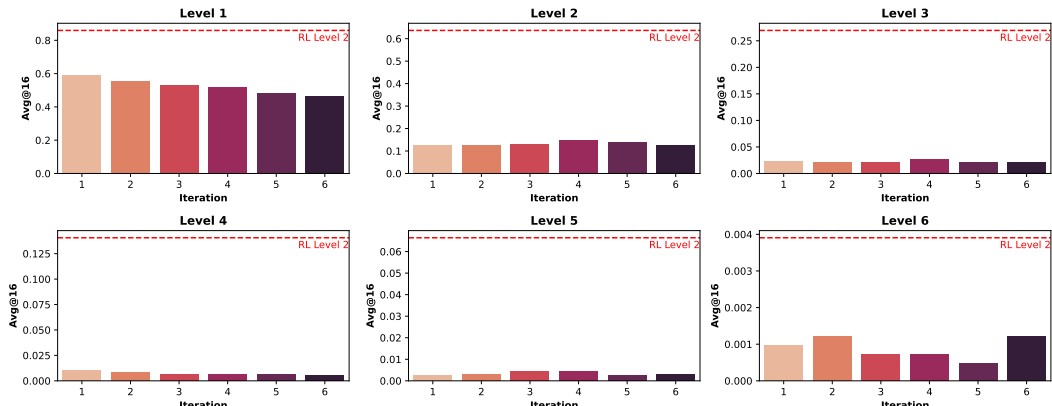

Figure 3: **RL vs. RFT on Compositional Tasks.** RL (red dashed line) achieves substantially higher accuracy across all levels, while iterative RFT fails to learn a generalizable skill.

Table 1: Model configurations for the task transfer experiment.

| Model Configuration | Stage 1 | | Stage 2 | |
| --- | --- | --- | --- | --- |
| | String Atomic RFT | Countdown Atomic RFT | String Atomic RL | String Comp. RL |
| String-Base + RL L1+2 | ✓ | ✗ | ✗ | ✓ |
| Multi-Base | ✓ | ✓ | ✗ | ✗ |
| Multi-Base + RL L1 | ✓ | ✓ | ✓ | ✗ |
| Multi-Base + RL L1+2 | ✓ | ✓ | ✗ | ✓ |

**Experimental Setup.** We test this conjecture on the Countdown task, where a model must construct a mathematical expression from a given set of integers to reach a target number (see §E for examples). In Countdown, a Level $\ell$ task requires the model to construct a mathematical expression using $\ell$ given integers to reach a target number. The minimum level for Countdown is Level 2. We compare four models to test our hypothesis, as detailed in Tab. 1. These configurations allow us to compare a "atomic-skill-only" baseline (Multi-Base) against models with either transferred atomic RL (Multi-Base + RL L1) or transferred compositional RL (Multi-Base + RL L1+2), as well as a control model from §4.1 that has the compositional skill but lacks the necessary atomic knowledge of Countdown (String-Base + RL L1+2). Note that *none* of the models are trained on Countdown with RL in Stage 2, and are only trained on our string task.

We evaluate these models on unseen, more challenging Countdown problems (Levels 3-5). We report the Avg@32, the average accuracy across 32 responses sampled at temperature 1.0.

**Results.** The results in Fig. 4 provide clear evidence supporting our hypothesis. The String-Base + RL L1+2 model fails completely. The Multi-Base model achieves reasonable accuracy of approximately 17% at Level 3 but still struggles at higher levels. Multi-Base + RL L1 shows marginal improvement over Multi-Base, increasing accuracy to around 20% at Level 3, with the advantage diminishing on more complex problems. The Multi-Base + RL L1+2 model achieves surprisingly strong performance. It achieves a 35% accuracy at Level 3, outperforming the Multi-Base baseline by more than 18% accuracy. This advantage persists at higher complexities, reaching approximately 6% at Level 4, where other models largely fail and achieve near-zero accuracy. The results show that the compositional skill learned from string transformation transfers to countdown, acting as a *meta-skill* that enhances the use of the

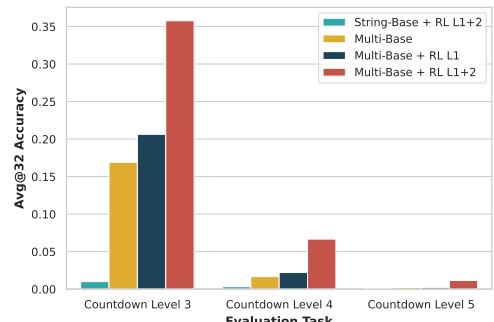

Figure 4: **Avg@32 Accuracy on the Countdown Task**. Atomic skills are a prerequisite for task transfer, and that compositional RL (Multi-Base + RL L1+2) on the unrelated string task offers a significant performance improvement on Countdown. Note that *none* of the models are trained with RL on Countdown.

target task's atomic knowledge. Finally, the comparison between Multi-base + RL and String-Base + RL L1+2 confirms our fundamental assumption that task-specific atomic skills are prerequisites for compositional skills to be effective.

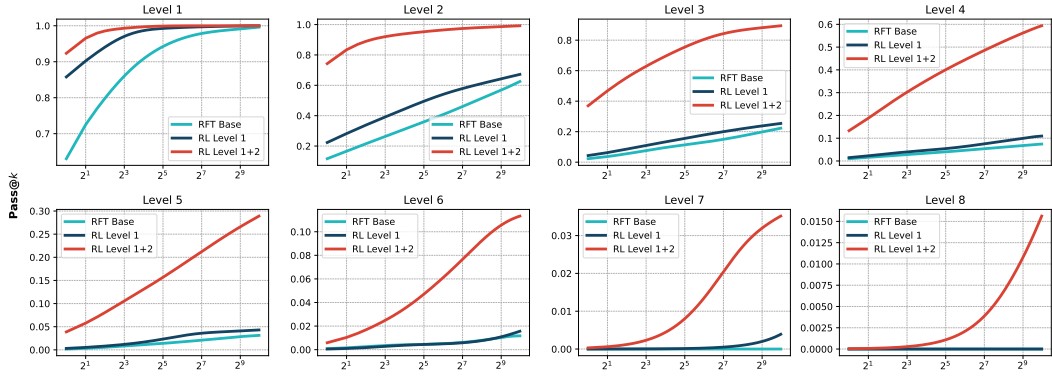

Figure 5: **Pass@$k$ performance across varying difficulty levels**. On easy problems (Levels 1-2), the performance gap shrinks with more samples, consistent with the *reranking* narrative. On hard problems (Levels 3-8), the gap widens substantially, suggesting new skill acquisition.

These results may explain recent findings on generalizable RL improvements. For example, Logic-RL (Xie et al., 2025) reports performance gains on mathematical problems after training on logic puzzles, and Guru (Cheng et al., 2025) shows that domains with greater pre-training exposure benefit more from cross-task generalization. We suggest that LLMs have already acquired essential atomic skills through large-scale pre-training, particularly in mathematics and coding. Thus, incentivizing compositional skills through RL in one task helps combine task-specific skills more effectively across domains. In contrast, domains with less pre-training exposure may lack sufficient atomic skills, limiting compositional skill transfer to downstream tasks. With this finding, we answer RQ3:

> **TAKEAWAY 3**
>
> Compositional skills learned through RL are transferable to a different task where the model possesses the atomic skills.

## 4.4 RL EXPANDING PERFORMANCE LIMITS IS NOT A FALSE PROMISE

Our findings strongly suggest that RL can teach compositional skills that are novel to the base model. This directly challenges recent arguments that RL merely "reranks" model responses, distilling pass@$k$ performance of the base model into pass@1 (Yue et al., 2025; Wu et al., 2025a). This conclusion is drawn based on a shrinking pass@$k$ performance gap between base and RL-tuned models as $k$ increases. However, we argue this conclusion may stem from two issues: (1) evaluating on mixed-skill benchmarks, therefore an improvement in a specific skill, like composition, can be masked in pass@$k$ if other required skills remain a bottleneck, and (2) using RL training that does not properly incentivize the new skill in the first place.

Our controlled framework allows us to dissect both issues. By isolating the compositional skill at varying difficulty levels, we can reliably assess skill acquisition (addressing issue 1), and by comparing different RL training setups (§4.1), we can test the effect of proper incentivization (addressing issue 2). We compare pass@1000 performance at each difficulty level of our test set, selecting $k = 1000$ as a sufficiently large and practically meaningful budget. Larger budgets would become impractical, as any reasonable model could theoretically achieve pass@$\infty = 1$.

The results are presented in Fig. 5. Both RL Level 1 and RL Level 1+2 models are trained from RFT base model using RL in Stage 2. The RL Level 1 model, which is *not* incentivized properly to learn composition, exhibits a similar trend to the RFT base across almost all levels. On easier problems (Levels 1 and 2) where the RFT base model already shows solving potential evidenced by high pass@k, the performance gaps between RL Level 1+2 model and the RFT model shrink as $k$ increases, aligning with the trends observed in Yue et al. (2025); Wu et al. (2025a). However, a completely different trend is observed on more challenging compositional problems (Levels 3-6). The RL Level 1+2 model's performance substantially outperforms the RFT base with an increasing gap as $k$ grows. For example, at Level 5, the performance gap over the RFT base grows from 4% at pass@1 to approximately 25% at pass@1024. This divergence is clear evidence of new skill acquisition. The results suggest that the pessimistic observation of "RL does not push performance

limits" in prior work may be explained by the lack of incentive for RL to learn new skills, as the base model already achieves high pass@k performance.

> **TAKEAWAY 4**
>
> The prior conclusion that RLVR only utilizes base models' reasoning patterns without learning new abilities is likely an artifact of evaluating and RL training on tasks that base models already achieve high pass@$k$; thus RL has little incentive to learn a new skill.

### 4.5 BEHAVIORAL ANALYSIS: RL TRANSFORMS FAILURE MODES

While our results show that training with compositional data unlocks promising generalization, a fundamental question remains: **do models trained under different setups exhibit different behaviors, or do they simply differ in capability while showing similar failure modes?** To investigate this, we analyze the failure modes of different models on Level 3 problems of our string task.

We use Gemini-2.5-Pro to classify responses into five categories: (1) **Correct**, (2) **Ignores Composition** (e.g., analyzing only a single function), (3) **Incomplete Trace** (recognizes composition but terminates early), (4) **Incorrect Composition** (e.g., misinterprets nesting), and (5) **Atomic Error** (errors in atomic functions prediction without the above). Categories 2-4 indicate difficulties with handling compositional problems. And while still incorrect, category 5 represents appropriate compositional behavior, as the error is not due to a lack of compositional skill.

We compare four models: **RFT Base** (after Stage 1 training), **RFT Level 2** (after Stage 2 training on Level 2 problems with RFT), **RL Level 1**, and **RL Level 2**, all from previous sections. The latter three models are all trained from the RFT Base.

Fig. 6 reveals substantial similarities in the failure patterns of RFT Base, RFT Level 2, and RL Level 1 models. Their failures are dominated by ignoring the composition entirely (all >50%) and misunderstanding the compositional structure (all >35%).

In contrast, the RL Level 2 model demonstrates fundamentally different behaviors. It completely eliminates "Ignores Composition" errors and correctly solves 28.1% of the problems. Crucially, its primary failure mode becomes "Atomic Error." This shows that compositional RL not only improves accuracy but teaches models to parse and execute compositional plans, shifting failures from high-level misunderstandings to lower-level execution errors. See §F for examples of different model responses.

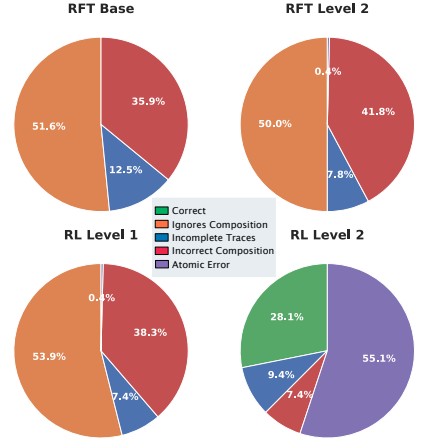

Figure 6: Distribution of failure modes on Level 3 string tasks. An "atomic error" is counted only when the model successfully resolves the compositional structure of the problem (i.e., it neither ignores the compositional operators nor misinterprets the nesting). Consequently, models that lack sufficient compositional capability do not have their atomic errors revealed or counted.

> **TAKEAWAY 5**
>
> Rather than merely improving accuracy, RL on compositional problems fundamentally transforms the model's behavior, enabling it to correctly understand and handle compositions.

## 5 CONCLUSION

The debate over whether RL can teach LLMs new skills has been clouded by experiments on benchmarks where LLMs already perform well, using coarse-grained metrics that obscure the learning of new capabilities. By stepping back to a cleaner, more controllable experimental environment, our findings provide a clear and optimistic answer: RL can teach genuinely new and powerful skills when the training task properly incentivize composition.

Our work focuses on the paradigm where RL acts upon pre-existing atomic competencies, mirroring the standard practice of carrying out RL based on pre-trained models. While our results show that possessing atomic skills is a sufficient condition for RL to unlock compositional capabilities, we do not claim it is strictly necessary. However, given the inefficiency of random exploration in discovering complex atomic behaviors from scratch, we argue that the synergy between supervised learning for atomic skill acquisition and RL for compositional generalization represents the most commonly adopted and promising path. Our findings suggest that the pessimistic conclusion that RL does not learn new skills may stem from inappropriate evaluation setups rather than fundamental constraints of RL itself. Future work may investigate the open question of whether RL can be scaled to acquire both atomic and compositional skills simultaneously without supervised scaffolding.

## LIMITATION

One limitation of this work is the reliance on synthetic tasks for evaluation. We deliberately designed our string transformation framework to enable rigorous experiments with well-defined atomic and compositional skills, thus isolating causal learning mechanisms. However, our synthetic tasks may not fully capture the complexity and nuance of real-world reasoning scenarios where skills are less clearly delineated and compositional structures are more varied. We acknowledge that demonstrating these findings in natural reasoning domains, such as mathematical problem-solving, code generation, or scientific reasoning, represents an important direction for future work. Extending these findings to realistic applications remains a valuable open challenge.

## ACKNOWLEDGEMENTS

This work is supported by NSF Grant No. CHE2505932, an Amazon AICE Award, gift funding from AI2, and a grant from Coefficient Giving.

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

## A    TRAINING DETAILS

### A.1    STAGE 1

All models in our experiments, except for the Multi-Base variants in §4.3, start from the same Stage 1 base model. The process for creating this model is detailed below.

**Data Generation.**   We first generate 50k Level-1 problems by randomly sampling a function and a string input (length 3 to 10). For each problem, we collect 10 responses from the `Llama-3.1-8B-Instruct` model (temperature 1.0, max length 8192). To focus training on problems the model finds non-trivial, we discard any problems where the base model achieves 100% accuracy. We then collect all remaining correct responses to form the SFT dataset. Before fine-tuning, we remove the function definitions from the prompts, keeping only the function identifier and the input. This results in around 116k training instances.

**Rejection Fine-Tuning.**   We fine-tune the model for 2 epochs with a learning rate of $2 \times 10^{-5}$ and a global batch size of 128. Other hyperparameters follow the default settings provided by the veRL framework[1].

### A.2    STAGE 2

**Data Generation.**   The problems used in Stage 2 are created using a similar process as in Stage 1, with the critical differences being that function definitions are hidden and problems can involve composition. For Level-2 problems, we randomly select and compose two functions. Example prompts for Stage 1 vs. Stage 2 can be found in Appendix C. For the Level 1 only and Level 2 only dataset, we create 50k problems. For the Level 1+2 mixed configuration, we create 25k problems for each level and combine them.

**Reinforcement Learning.**   For RL experiments, we use DAPO (Yu et al., 2025) as the optimization algorithm. We enforce a strictly on-policy setup by setting both the training batch size and mini-batch size to 16. For each prompt, we generate 16 rollouts using a sampling temperature of 1.0 and a maximum response length of 8192. During training, we filter out any problems for which all rollouts are correct or all are incorrect. We use a learning rate of $1 \times 10^{-6}$ and set the coefficients for both KL divergence and entropy loss to 0.

**Rejection Fine-Tuning.**   For the iterative RFT baseline, we use a learning rate of $2 \times 10^{-5}$ and a batch size of 128. The process is iterative: the model from the previous iteration is used to generate a new set of rollouts. These rollouts are then filtered for correctness to construct a new SFT dataset, following the same procedure used in Stage 1. The model is then trained on this new dataset to produce the model for the next iteration.

## B    EVALUATION DETAILS

**String Transformation Prediction Task.**   For the string transformation prediction task, the evaluation set consists of 256 randomly generated problems for each level from 1 to 8, resulting in a total of 2048 test problems. During evaluation, we generate responses using a sampling temperature of 1.0 and a maximum response length of 8192 tokens.

**Countdown Task.**   The evaluation set for the Countdown task consists of 128 problems for each level. The evaluation setup is the same as evaluating on the string transformation prediction task.

## C    EXAMPLE PROMPTS FOR STAGE 1 AND STAGE 2

Here we present the example prompts used in our experiments. Fig. 7 shows the prompt used for Stage 1 data generation, where the function definition is provided to guide the model. However,

---

[1]`https://github.com/volcengine/verl`

```
You are given a code:

def func_16(s):
    """Remove adjacent duplicate characters (compress repeats)."""
    if not s:
        return s
    result = [s[0]]
    for ch in s[1:]:
        if ch != result[-1]:
            result.append(ch)
    return ''.join(result)

def main_solution(x):
    return func_16(x)

Can you predict the output of 'main_solution("tihess")' without writing any code? Please reason and put
 your final answer in the following json format: {"output": <your output>}, where <your output> should
be the final string.
```

Figure 7: Example prompt for Stage 1 response collection. Note that here the function definition is given, but during Stage 1 RFT, the function definition is removed.

```
Example prompt for Stage 1 and Stage 2 Level 1 training
You are given a code:

def main_solution(x):
    return func_16(x)

Can you predict the output of 'main_solution("tiheass")' without writing any code? Please reason and
put your final answer in the following json format: {"output": <your output>}, where <your output>
should be the final string.

Example prompt for Stage 2 Level 2 training
You are given a code:

def main_solution(x):
    return func_2(func_16(x), 3)

Can you predict the output of 'main_solution("tiheass")' without writing any code? Please reason and
put your final answer in the following json format: {"output": <your output>}, where <your output>
should be the final string.
```

Figure 8: Example prompts for evaluation and Stage 2 training.

during the actual Stage 1 training (and throughout Stage 2), this definition is removed, forcing the model to rely on the internalized atomic skills associated with the function identifier. Fig. 8 shows the prompt used for Stage 1 and Stage 2 training.

## D  A COMPLETE LIST OF STRING TRANSFORMATION FUNCTIONS

Here we provide the full list of string functions we use. Note that we replace the function names with meaningless identifiers such as `func_1` in our experiment.

```python
def deterministic_shuffle(s):
    """Reorder characters using a fixed multiplier permutation."""
    L = len(s)
    if L == 0:
        return s
    multiplier = 3
    while gcd(multiplier, L) != 1:
        multiplier += 2
    return ''.join(s[(i * multiplier) % L] for i in range(L))

def repeat_str(s, n):
    """Repeat the string s exactly n times."""
    return s * n

def remove_vowels(s):
    """Remove vowels from the string."""
    vowels = 'aeiouAEIOU'
    return ''.join(ch for ch in s if ch not in vowels)

def sort_chars(s):
    """Sort the characters in the string."""
```

```python
    return ''.join(sorted(s))

def reverse_words(s):
    """Reverse the order of words in the string."""
    words = s.split()
    return ' '.join(reversed(words))

def add_prefix(s, pre):
    """Add a fixed prefix to the string."""
    return pre + s

def add_suffix(s, suf):
    """Add a fixed suffix to the string."""
    return s + suf

def interlace_str(s1, s2):
    """Interlace two strings character by character (iterative)."""
    result = []
    len1, len2 = len(s1), len(s2)
    for i in range(max(len1, len2)):
        if i < len1:
            result.append(s1[i])
        if i < len2:
            result.append(s2[i])
    return ''.join(result)

def rotate_str(s, n):
    """Rotate the string s by n positions using slicing."""
    if not s:
        return s
    n = n % len(s)
    return s[n:] + s[:n]

def mirror_str(s):
    """Append the reversed string to the original."""
    return s + s[::-1]

def alternate_case(s):
    """Alternate the case of characters (even-index lower, odd-index
    upper)."""
    return ''.join(ch.lower() if i % 2 == 0 else ch.upper() for i, ch in
     enumerate(s))

def shift_chars(s, shift):
    """
    Shift alphabetical characters by a fixed amount (wrapping around).
    Non-letters remain unchanged.
    """

    def shift_char(ch):
        if 'a' <= ch <= 'z':
            return chr((ord(ch) - ord('a') + shift) % 26 + ord('a'))
        elif 'A' <= ch <= 'Z':
            return chr((ord(ch) - ord('A') + shift) % 26 + ord('A'))
        return ch

    return ''.join(shift_char(ch) for ch in s)
```

```python
def vowel_to_number(s):
    """Replace vowels with numbers: a/A->1, e/E->2, i/I->3, o/O->4, u/U
    ->5."""
    mapping = {'a': '1', 'e': '2', 'i': '3', 'o': '4', 'u': '5', 'A': '1
    ', 'E': '2', 'I': '3', 'O': '4', 'U': '5'}
    return ''.join(mapping.get(ch, ch) for ch in s)

def insert_separator(s, sep):
    """Insert a fixed separator between every two characters."""
    return sep.join(s)

def duplicate_every_char(s):
    """Duplicate every character in the string."""
    return ''.join(ch * 2 for ch in s)

def fancy_brackets(s):
    """Enclose each character in fancy brackets."""
    return ''.join("<<" + ch + ">>" for ch in s)

def compress_repeats(s):
    """Remove adjacent duplicate characters (compress repeats)."""
    if not s:
        return s
    result = [s[0]]
    for ch in s[1:]:
        if ch != result[-1]:
            result.append(ch)
    return ''.join(result)

def recursive_reverse(s):
    """Recursively reverse the string."""
    if s == "":
        return s
    return recursive_reverse(s[1:]) + s[0]

def loop_concat(s, n):
    """Concatenate s with itself n times using a loop."""
    result = ""
    for _ in range(n):
        result += s
    return result

def while_rotate(s, n):
    """Rotate the string using a while loop (n times)."""
    count = 0
    while count < n and s:
        s = s[1:] + s[0]
        count += 1
    return s

def recursive_interlace(s1, s2):
    """Recursively interlace two strings character by character."""
    if not s1 or not s2:
        return s1 + s2
    return s1[0] + s2[0] + recursive_interlace(s1[1:], s2[1:])
```

```python
def loop_filter_nonalpha(s):
    """Remove non-alphabetic characters using an explicit loop."""
    result = ""
    for ch in s:
        if ch.isalpha():
            result += ch
    return result

def verify_even_length(s):
    """
    Verification operator: if the length of s is even, return s;
    otherwise remove the last character.
    """
    return s if len(s) % 2 == 0 else s[:-1]

def backchain_add_digit(s, depth):
    """
    Backtracking operator: deterministically transform s so it contains
    a digit.
    Applies a fixed sequence of transformations recursively.
    """

    def has_digit(t):
        return any(ch.isdigit() for ch in t)

    transformations = [
        lambda t: t + "1",
        lambda t: "2" + t,
        lambda t: t.replace("a", "3"),
        lambda t: t[::-1],
    ]

    def helper(t, d):
        if has_digit(t):
            return t
        if d == 0:
            return None
        for trans in transformations:
            new_t = trans(t)
            res = helper(new_t, d - 1)
            if res is not None:
                return res
        return None

    result = helper(s, depth)
    return result if result is not None else s

def backchain_palindrome(s, depth):
    """
    Back chaining: try to transform s into a palindrome.
    If s is not already a palindrome and depth permits, append its
    reverse and try again.
    """
    if s == s[::-1]:
        return s
    if depth <= 0:
        return s
    new_s = s + s[::-1]
    return backchain_palindrome(new_s, depth - 1)

# [note: the string concatenation, i.e., '+' is also considered as a
function]
```

---

Listing 1: All functions

---

# E    EXAMPLE FOR COUNTDOWN TASK

```
Level 2 Example
Using the numbers [32, 42], create an equation that equals 74. You can use basic arithmetic operations
(+, -, *, /) and each number can only be used once. Show your work in <think> </think> tags. And return
 the final answer in <answer> </answer> tags, for example <answer> (1 + 2) / 3 * 4 </answer>.

Level 3 Example
Using the numbers [95, 14, 18], create an equation that equals 99. You can use basic arithmetic
operations (+, -, *, /) and each number can only be used once. Show your work in <think> </think> tags.
 And return the final answer in <answer> </answer> tags, for example <answer> (1 + 2) / 3 * 4 </answer
>.
```

Figure 9: Example prompts for Countdown task.

# F    MODEL RESPONSE EXAMPLES

The following sections contain the complete, unfolded responses from each model for the case studies.

## F.1    FULL RESPONSES TO THE LEVEL 2 PROBLEM

---

### RFT Model Response

To predict the output of 'main_solution("nar")', we need to understand how the function 'func_7' interlaces two input strings. *[Authors' Note: It's not wrong, but it misses another function func_2.]*

Here's a step-by-step breakdown:

1. 'func_7(s1, s2)' is called with ''vptqj'' as the first string 's1' and '"nar"' as the second string 's2'.
2. The function iterates over the maximum length between the two input strings. In this case, the maximum length is 4, which is the length of '"nar"'.
3. The function appends characters from both strings at each index 'i' to the 'result' list. If an index is out of range for a particular string, it simply skips appending a character from that string.
4. So, the 'result' list will contain characters from both strings, interleaved.

Here's the interleaved sequence:

- Index 0: ''v'' (from ''vptqj'') and ''n'' (from '"nar"'), so 'result = ['v', 'n']'
- Index 1: ''p'' (from ''vptqj'') and ''a'' (from '"nar"'), so 'result = ['v', 'n', 'p', 'a']'
- Index 2: ''t'' (from ''vptqj'') and ''r'' (from '"nar"'), so 'result = ['v', 'n', 'p', 'a', 't', 'r']'
- Index 3: ''q'' (from ''vptqj'') and '' '' (from '"nar"'), so 'result = ['v', 'n', 'p', 'a', 't', 'r', 'q',"]

Finally, we pass this 'result' list to the 'join()' function to form the final interleaved string.

The final output of 'main_solution("nar")' will be: '"vnpartqj"'. *[Authors' Note: End too early, there's still one function that hasn't been analyzed.]*

So, the JSON output should be:

```
{"output": "vnpartqj"}
```

---

---

**RL Level 1 Model Response**

To predict the output of 'main_solution("nar")', we need to understand how the function 'func_7' interlaces two strings character by character.

Here's a step-by-step breakdown:

1. 'func_7' takes two strings 's1' and 's2' as input.
2. It creates an empty list 'result' to store the interlaced characters.
3. It calculates the lengths of 's1' and 's2' and stores them in 'len1' and 'len2', respectively.
4. It loops from '0' to the maximum of 'len1' and 'len2' using a 'for' loop.
5. Inside the loop, for each iteration 'i', it checks if 'i' is less than 'len1' and 'len2'. If it is, it appends the 'i-th' character of 's1' and 's2' to the 'result' list, if it exists.
6. After the loop, it returns the 'result' list as a string by joining all its elements using '".join(result)'.

Now, let's analyze the 'main_solution' function:

'def main_solution(x): return (func_7('vptqj', x) + func_2('xbh'))'

Here, 's1' is ''vptqj'' and 's2' is 'x', which is '"nar"' in this case.

Now, let's interlace the two strings character by character:

'i |s1 |s2 |result'
–|—-|—-|———
0 |'v' |'n' |['v', 'n']
1 |'p' |'a' |['v', 'n', 'p', 'a']
2 |'t' |'r' |['v', 'n', 'p', 'a', 't', 'r']
3 |'q' ||['v', 'n', 'p', 'a', 't', 'r', 'q']
4 |'j' ||['v', 'n', 'p', 'a', 't', 'r', 'q', 'j']
5 |||['v', 'n', 'p', 'a', 't', 'r', 'q', 'j']
6 |||['v', 'n', 'p', 'a', 't', 'r', 'q', 'j']

The loop iterates up to the maximum of the lengths of 's1' and 's2', which is 5 in this case. After the loop, 'func_7' returns the 'result' list as a string by joining all its elements.

The 'result' list is '['v', 'n', 'p', 'a', 't', 'r', 'q', 'j']'.

The 'func_2' function is used to remove vowels from a string. It takes a string 's' as input and returns a new string with all vowels removed.

In the 'main_solution' function, 'func_2' is used to remove vowels from the string ''xbh'', but the output of 'func_7' is used as the input for 'func_2' *[Authors' Note: the model mistakes the relationship between the two functions]*. However, in the 'main_solution' function, 'func_2' is not actually used with the interlaced string but with the string ''xbh''.

The 'main_solution' function returns the result of 'func_7('vptqj', 'nar') + func_2('xbh')'.

Since 'func_2' is not used with the interlaced string, the output of 'main_solution("nar")' will be the interlaced string '['v', 'n', 'p', 'a', 't', 'r', 'q', 'j']' as a string, which is:

'vnpatrqj'

So, the output of 'main_solution("nar")' is:

```
{"output": "vnpatrqj"}
```

**RL Level 2 Model Response**

To predict the output of 'main_solution("nar")', we need to understand what the function 'func_7' does and how it is used in 'main_solution'.

'func_7' interlaces two strings character by character. It takes two strings 's1' and 's2' as input and returns a new string where characters from 's1' and 's2' are alternately appended.

In 'main_solution', 'func_7' is called with ''vptqj'' as the first string and the input 'x' as the second string. The result of 'func_7' is then concatenated with the result of 'func_2('xbh')'.

Now, let's consider what happens when 'main_solution' is called with '"nar"' as the input.

First, we need to call 'func_7' with ''vptqj'' and '"nar"' as the two strings.

The characters in ''vptqj'' are ''v', 'p', 't', 'q', 'j''.

The characters in '"nar"' are ''n', 'a', 'r''.

Now, let's interlace these characters using 'func_7'.

The maximum length of the two strings is 5. So, 'func_7' will iterate up to 5 times.

Here's what happens during each iteration:

1. 'i = 0': 's1[i] = 'v'' and 's2[i] = 'n''. The result list is '['v', 'n']'.
2. 'i = 1': 's1[i] = 'p'' and 's2[i] = 'a''. The result list is '['v', 'n', 'p', 'a']'.
3. 'i = 2': 's1[i] = 't'' and 's2[i] = 'r''. The result list is '['v', 'n', 'p', 'a', 't', 'r']'.
4. 'i = 3': 's1[i] = 'q'' and 's2[i] = '''. The result list is '['v', 'n', 'p', 'a', 't', 'r', 'q']'.
5. 'i = 4': 's1[i] = 'j'' and 's2[i] = '''. The result list is '['v', 'n', 'p', 'a', 't', 'r', 'q', 'j']'.

The final result list after iterating over all characters from both strings is '['v', 'n', 'p', 'a', 't', 'r', 'q', 'j']'.

Now, we need to find the characters in ''xbh'' and remove them from 'func_2'. The characters in ''xbh'' are ''x', 'b', 'h''.

Now, let's consider what 'func_2' does. This function removes all vowels from a string.

The input string to 'func_2' is ''xbh''. The vowels in ''xbh'' are '''' (there are no vowels in ''xbh'').

However, we need to remove all vowels from ''xbh''. Since there are no vowels in ''xbh'', the string remains the same.

The function 'func_2' will return ''xbh'' as is.

Now, let's consider what happens when we concatenate the result from 'func_7('vptqj', "nar")' with 'func_2('xbh')'. As explained earlier, 'func_7('vptqj', "nar")' will return ''vnpatrqj''.

Now, since 'func_2('xbh')' just returns ''xbh'', the final output of 'main_solution("nar")' will be the concatenation of ''vnpatrqj'' and ''xbh''.

The concatenation of ''vnpatrqj'' and ''xbh'' is ''vnpatrqjxbh''.

So, the final output of 'main_solution("nar")' is ''vnpatrqjxbh''.

Here's the JSON output:

```
{"output": "vnpatrqjxbh"}
```

## F.2  FULL RESPONSES TO THE LEVEL 3 PROBLEM

### RFT Model Response

To predict the output of 'main_solution("htoek")', let's analyze the code.

1. The 'main_solution' function calls 'func_14' with the input string 'x'.
2. 'func_14' takes a string 's' and returns a new string where each character's case is alternated based on its index.
- Even-index characters (i.e., those at 0, 2, 4, etc. indices) are converted to lowercase.
- Odd-index characters (i.e., those at 1, 3, 5, etc. indices) are converted to uppercase.
3. The input string 'x' to 'main_solution' is '"htoek"'.

Now, let's apply the logic of 'func_14' to the input string '"htoek"':

- Index 0 is even, so 'h' is converted to lowercase: 'h'.
- Index 1 is odd, so 't' is converted to uppercase: 'T'.
- Index 2 is even, so 'o' is converted to lowercase: 'o'.
- Index 3 is odd, so 'e' is converted to uppercase: 'E'.
- Index 4 is even, so 'k' is converted to lowercase: 'k'.

The resulting string is ''hToEk''.

Therefore, the output of 'main_solution("htoek")' is: *[Authors' Note: Ends too early, there's still one function that hasn't been analyzed]*

```
{"output": "hToEk"}
```

### RL Level 1 Model Response

To predict the output of 'main_solution("htoek")', we need to understand what the 'func_10' function does and how it is used in 'main_solution'.

'func_10' takes a string 's' and returns a new string where:

* Even-indexed characters are converted to lowercase
* Odd-indexed characters are converted to uppercase

'main_solution' calls 'func_10' with the string 'x' after applying the 'func_2' function, which removes all vowels from 'x'. *[Authors' Note: the model fails to correctly understand and parse the nesting function]*

Now, let's apply these steps:

1. 'func_2' removes vowels from '"htoek"', resulting in '"htk"'.
2. 'func_10' is applied to '"htk"'.

- 'h' is at index 0 (even), so it becomes 'h' (lowercase).
- 't' is at index 1 (odd), so it becomes 'T' (uppercase).
- 'o' is at index 2 (even), so it becomes 'o' (lowercase).
- 'e' is at index 3 (odd), so it becomes 'E' (uppercase).
- 'k' is at index 4 (even), so it becomes 'k' (lowercase).

The resulting string is '"hToEk"'.

However, the question asks us to predict the output without writing any code, so we might have mistakenly included the characters from the original string '"htoek"' instead of the output of 'func_2'.

To correctly predict the output, we should first apply 'func_2' to '"htoek"'. The vowels in '"htoek"' are "o" and "e". Removing these vowels leaves us with '"htk"'.

Now, applying 'func_10' to '"htk"':
- 'h' is at index 0 (even), so it becomes 'h' (lowercase).
- 't' is at index 1 (odd), so it becomes 'T' (uppercase).
- 'k' is at index 2 (even), so it becomes 'k' (lowercase).

The resulting string is '"hTk"'.

This is the output of 'func_10("htk")', which is what 'main_solution' returns.

Therefore, the output of 'main_solution("htoek")' is '"hTk"'.

The final answer is:

```
{"output": "hTk"}
```

---

**RL Level 2 Model Response**

To predict the output of 'main_solution("htoek")', let's analyze the code.

1. The 'func_2' function removes all vowels from a given string 's'.
2. The 'func_14' function takes a string 's' and returns a new string where every character in 's' is duplicated.
3. The 'func_10' function iterates over the characters in the input string 's'. It makes the characters at even indices lower case and the characters at odd indices upper case.

Now, let's see what happens when we call 'main_solution("htoek")'.

First, we need to find the output of 'func_2("htoek")'.

The string "htoek" contains the characters "h", "t", "o", "e", "k". All characters that are vowels (i.e., "o" and "e") will be removed from the string.

After removing vowels, the string 'func_2("htoek")' will contain only the characters "h", "t", "k".

Now, we need to find the output of 'func_14("htoek")'.

The characters in "htoek" are "h", "t", "o", "e", "k".

Applying 'func_14' to each character:
- "h" becomes "hh"

- "t" becomes "tt"
- "o" becomes "oo"
- "e" becomes "ee"
- "k" becomes "kk"

The string 'func_14("htoek")' will be "hhttooeekk".

Now, we need to find the output of 'func_10("hhttooeekk")'.

We will iterate over the characters in "hhttooeekk" along with their indices.

The characters in "hhttooeekk" and their indices are:
- "h" at index 0
- "h" at index 1
- "t" at index 2
- "t" at index 3
- "o" at index 4
- "o" at index 5
- "e" at index 6
- "e" at index 7
- "k" at index 8
- "k" at index 9

Now, let's apply the function 'func_10' to "hhttooeekk".

- The character at index 0 ("h") will become lower case because the index is even. So, "h" becomes "h".
- The character at index 1 ("h") will become upper case because the index is odd. So, "h" becomes "H".
- The character at index 2 ("t") will become lower case because the index is even. So, "t" becomes "t".
- The character at index 3 ("t") will become upper case because the index is odd. So, "t" becomes "T".
- The character at index 4 ("o") will become lower case because the index is even. So, "o" becomes "o".
- The character at index 5 ("o") will become upper case because the index is odd. So, "o" becomes "O".
- The character at index 6 ("e") will become lower case because the index is even. So, "e" becomes "e".
- The character at index 7 ("e") will become upper case because the index is odd. So, "e" becomes "E".
- The character at index 8 ("k") will become lower case because the index is even. So, "k" becomes "k".
- The character at index 9 ("k") will become upper case because the index is odd. So, "k" becomes "K".

The final string 'func_10("hhttooeekk")' will be "hHtToOeEkK".

Now, let's call 'main_solution("htoek")'.

Since 'main_solution(x)' calls 'func_10(func_14(func_2(x)))', we need to find the output of 'func_14(func_2("htoek"))' first.

We already found that 'func_2("htoek")' will contain only the characters "h", "t", "k".

Now, let's apply 'func_14' to "htk".

The characters in "htk" are "h", "t", "k".

Applying 'func_14' to each character:
- "h" becomes "hh"
- "t" becomes "tt"
- "k" becomes "kk"

The string 'func_14("htk")' will be "hhttkk".

Now, we need to find the output of 'func_10("hhttkk")'.

The characters in "hhttkk" and their indices are:
- "h" at index 0
- "h" at index 1
- "t" at index 2
- "t" at index 3
- "k" at index 4
- "k" at index 5

Now, let's apply the function 'func_10' to "hhttkk".

- The character at index 0 ("h") will become lower case because the index is even. So, "h" becomes "h".
- The character at index 1 ("h") will become upper case because the index is odd. So, "h" becomes "H".
- The character at index 2 ("t") will become lower case because the index is even. So, "t" becomes "t".
- The character at index 3 ("t") will become upper case because the index is odd. So, "t" becomes "T".
- The character at index 4 ("k") will become lower case because the index is even. So, "k" becomes "k".
- The character at index 5 ("k") will become upper case because the index is odd. So, "k" becomes "K".

The string 'func_10("hhttkk")' will be "hHtTkK".

Now, we need to call 'main_solution("htoek")'.

As 'main_solution(x)' calls 'func_10(func_14(func_2(x)))', the output of 'main_solution("htoek")' will be the output of 'func_10(func_14(func_2("htoek")))'.

We already found that 'func_14("hhttooeekk")' will be "hHtToOeEkK".

Now, we need to find the output of 'func_10("hHtToOeEkK")'.

We already calculated this in the previous step. The output of 'func_10("hHtToOeEkK")' will be "hHtToOeEkK".

However, we need to recall that 'func_2' removes vowels from the string. So, to find the output of 'func_10(func_14(func_2("htoek")])', we need to first find the output of 'func_14(func_2("htoek"))'.

We already calculated this earlier. The output of 'func_14(func_2("htoek"))' will be "hhttkk".

Now, we need to find the output of 'func_10("hhttkk")'.

We already calculated this earlier. The output of 'func_10("hhttkk")' will be "hHt-TkK".

The 'main_solution' function calls 'func_10(func_14(func_2(x)))'. To find the final output of 'main_solution("htoek")', we need to find the output of 'func_14(func_2("htoek"))'.

The string "htoek" has characters "h", "t", "o", "e", "k". The vowels in this string are "o" and "e".

If we remove these vowels from "htoek", we are left with "htk".

Now, let's apply 'func_14' to "htk".

The characters in "htk" are "h", "t", "k".

Applying 'func_14' to each character:
- "h" becomes "hh"
- "t" becomes "tt"
- "k" becomes "kk"

The string 'func_14("htk")' will be "hhttkk".

Now, let's apply 'func_10' to "hhttkk".

We already calculated this earlier. The output of 'func_10("hhttkk")' will be "hHt-TkK".

The 'main_solution' function calls 'func_10(func_14(func_2(x)))'. To find the final output of 'main_solution("htoek")', we need to find the output of 'func_14(func_2("htoek"))'.

The characters in "htoek" are "h", "t", "o", "e", "k". The vowels in this string are "o" and "e".

If we remove these vowels from "htoek", we are left with "htk".

Now, let's apply 'func_14' to "htk".

We already calculated this earlier. The output of 'func_14("htk")' will be "hhttkk".

Now, we can apply 'func_10' to "hhttkk".

We already calculated this earlier. The output of 'func_10("hhttkk")' will be "hHt-TkK".

So, the output of 'main_solution("htoek")' will be "hHtTkK".

The final answer is:

```
{"output": "hHtTkK"}
```

