# OpenReview forum: "From f(x) and g(x) to f(g(x)): LLMs Learn New Skills in RL by Composing Old Ones"
_ICLR.cc/2026/Conference — ICLR 2026 Poster_

### Official Review · Reviewer_UU3i · 2025-10-30

**Soundness:** 3
**Presentation:** 4
**Contribution:** 3
**Rating:** 6
**Confidence:** 4

**Summary:**

This paper argues that LLMs can acquire genuinely new skills during RL by composing existing ones.
Using a controllable synthetic framework, this paper shows that LLMs compose learned string transformation functions via RL, and such compositional capability can transfer to different tasks. The paper also identifies that the key is to do RL on compositional data which incentivizes the compositionality.

**Strengths:**

- Studying whether RL geniunely teaches new skills is a very timely topic.

- The controllable framework plus decontamination procedure enhances the soundness of the conclusion.

- The paper is well-written and easy to read.

**Weaknesses:**

1. While the the paper provides a good controllable framework to understand what's happening during RL, I wonder what the actionable point is for future works. For example, Takeaway 3 implies that, one could hand-design (or synthesize) some compositional samples manually and controllably (just like the string transformation functions proposed), and train on those samples to incentivize a better compositional reasoning in more practical domains (than the countdown task) like code or math. I hope the authors can comment on this front or add some experimental results (even preliminary) in the paper.

2. The paper claims that RL on compositional data can teach compositional skills, 'if the atomic skills are already learned'. It's not clear that whether the condition is necessary. What if training RL directly on a mixture of atomic data and compositional data (thus without the stage 1)? Would RL be able to learn atomic and compositional skills together? I understand this might make experimental setup more complicated, such as leaking the function definitions, but maybe good to add a discussion on this.

**Questions:**

1. I wonder how is the conclusions connect with the enropy/diversity view of RLVR?


2. What is the specific reason of setting the coefficients for both KL divergence and entropy loss to 0?


3. How do you ensure that RL and RFT in stage 2 are in a fair comparison? How are the hyperparameters selected, do both see a same amount of training samples? These questions seemed not to be reported.

---

> ### Author Response · Authors · 2025-11-21
> **Response to Reviewer UU3i (1/2)**
>
> We thank you for your positive and constructive review. You've raised several excellent points and questions, which we are happy to address.
>
> ---
>
> ### On Weakness 1: Actionable Points for Future Work
>
> Thanks for the suggestion. We believe that our paper provides strong evidence to show that RL teaches LLMs to learn new skills, despite the overwhelming support of the opposite view [1] (The paper received the only full score in NeurIPS 2025 and triggered a heated discussion). Therefore, the first and foremost “actionable point” is to encourage the community to keep investing in RL.
>
> Secondly and more technically, the "actionable point" of our paper is a **general principle** for approaching complex skill acquisition: **"learn the atoms, then compose."**
>
> Our work suggests that for any new, complex domain, future work could benefit from:
>
> 1. **Learning Atomic Skills:** First, try to decompose the complex task into its non-trivial "atomic" components. For example, in a **visual CoT** task, the ability to "use tool" (like cropping, zooming, etc.) and the ability to reason could be seen as two separate atomic skills (although they are of a higher level than the atomic skills defined in our paper). For **agent coding** tasks, the generation of reusable functions or classes could act as the "atoms."
> 2. **Training Atomic Skills:** Ensure the model has mastered these atomic skills first, perhaps through targeted pre-training or supervised fine-tuning (our "Stage 1").
> 3. **Incentivizing Composition:** Finally, use RL on tasks that *require the composition* of these atoms (maybe simply do the end-to-end RL on the task that requires these atomic skills) to teach the model the *generalizable skill* of composition itself.
>
> We believe this principle could lead to faster convergence and better generalization in many practical domains. Thank you for highlighting this key implication, and we will add the discussion to our paper.
>
> ---
>
> ### On Weakness 2: Is Stage 1 Strictly Necessary?
>
> Thanks for the question. Our paper shows that *when* a model has a non-trivial level of atomic skill, proper RL *can* incentivize composition. We acknowledge that we do not prove that this is the *only* path. This is a sufficient but not necessarily a necessary condition.
>
> However, we designed our experiment to be a reasonable simulation of real-world scenarios, where atomic skills are typically learned during pre-training or SFT (our Stage 1), not from scratch with RL. It is commonly recognized that RL is often not an efficient way to teach a brand-new atomic skill if that skill is not already present (i.e., discoverable) in the model's policy. Relying on random rollouts to discover a complex atomic function's behavior would be highly inefficient and perhaps impossible.
>
> Training RL directly on a mixture of atomic and compositional data (without our Stage 1) would also complicate the experimental setup. As you note, we would have to provide the function definitions, at which point the "atomic skill" is no longer the one we are trying to teach, but rather the model's pre-existing coding ability (because it now relies on the model’s understanding of the code, the model can do in-context inference, which is learned during pre-training and post-training). So if we provide the function definitions, then the model’s original pre-training and post-training act as the “stage 1 training” in our paper.
>
> Additionally, Figure 4 shows that directly transferring a model trained with compositional RL solely on the string task to the Countdown task yields poor performance (see the **String-Base** bar). A non-trivial performance gain from RL only emerges once the model has already acquired the **atomic Countdown skill**. From this perspective, Stage 1 appears to be necessary.
>
> We believe the question of whether RL can *simultaneously* learn atomic and compositional skills from scratch is another important research question, but it is one that our current work, which focuses on *composing existing skills,* leaves for future investigation.
>
> We have added a discussion in our conclusion section, and have uploaded the revised manuscript. Again, thank you for your constructive comment.

---

> > ### Author Response · Authors · 2025-11-21
> > **Response to Reviewer UU3i (2/2)**
> >
> > ### On Question 1: Connection to Entropy/Diversity View
> >
> > Thanks for the question. In our own analysis, we did not find a direct connection. We observed that the entropy in both our `RL Level 1` and `RL Level 1+2` runs showed a similar decreasing trend during training. This suggests that entropy might be a lower-level metric that does not fully capture the high-level *strategic* differences between the models. Our `RL Level 1+2` model learned a fundamentally new compositional strategy (Figure 6), even while its entropy trajectory looked similar to the model that failed to learn this skill. Therefore, we call for better metrics to reflect whether a model learns new skills during RL.
> >
> > ---
> >
> > ### On Question 2: KL and Entropy Loss Coefficients
> >
> > We set the coefficients for both KL divergence and entropy loss to 0, which is a common practice in many recent works on RL for LLMs. We followed this established setup [2,3].
> >
> > ---
> >
> > ### On Question 3: Fair Comparison between RL and RFT
> >
> > We took several steps to ensure a fair comparison between our Stage 2 RL and RFT baselines:
> >
> > - **Identical Data Generation:** Both methods started from the exact same Stage 1 base model. They were then trained on problems sampled from the same distribution, using the same prompts.
> > - **Identical Sampling:** We used the same model sampling parameters (temperature, max length, etc.) and the same number of rollouts per prompt during data generation for both setups. This means both methods "saw" a roughly equivalent amount of training data.
> > - **Algorithmic Difference:** The primary difference is how the collected data is used. RFT discards all negative (incorrect) rollouts. Our RL algorithm (GRPO), on the other hand, can leverage negative rollouts as long as there is at least one positive rollout for that prompt. This is an *inherent* difference between the learning paradigms, but we believe the comparison is as fair as it can be, as the inputs and data-gathering budget were matched.
> >
> > [1] Yue, Yang, et al. "Does reinforcement learning really incentivize reasoning capacity in llms beyond the base model?, 2025." *URL https://arxiv. org/abs/2504.13837*.
> >
> > [2] Yu, Qiying, et al. "Dapo: An open-source llm reinforcement learning system at scale." *arXiv preprint arXiv:2503.14476* (2025).
> >
> > [3] Shao, Rulin, et al. "Spurious rewards: Rethinking training signals in rlvr." *arXiv preprint arXiv:2506.10947* (2025).

---

> > > ### Comment · Reviewer_UU3i · 2025-11-23
> > >
> > > I appreciate the authors for their concise response. I do think adding the discussion on stage 1 at the end is useful (it helps tone down the claims). I'll maintain my score.

---

### Official Review · Reviewer_xBeE · 2025-10-31

**Soundness:** 3
**Presentation:** 3
**Contribution:** 3
**Rating:** 6
**Confidence:** 3

**Summary:**

This paper investigates whether reinforcement learning (RL) genuinely teaches new reasoning skills to large language models (LLMs). Using a synthetic, contamination-free string transformation benchmark, the authors define “atomic” and “compositional” skills and show that RL with compositional data enables models to generalize to deeper compositions unseen during training. In contrast, rejection fine-tuning (RFT) or RL on atomic data fails to do so. The learned skill also transfers modestly to a different task (Countdown) when atomic skills are present. Behavioral analyses reveal that RL changes reasoning patterns rather than merely reranking outputs. Overall, the paper argues that RL can create new compositional reasoning abilities if properly incentivized.

**Strengths:**

The question (“does RL truly teach new skills?”) is central and timely.

The synthetic setup is impressively controlled and eliminates confounders such as data contamination.

Clear evidence that RL + compositional data leads to new behaviors unseen in supervised baselines.

Strong writing quality and interpretability of experiments (the “Takeaway” summaries are exemplary).

**Weaknesses:**

The paper’s analysis mostly rely on synthetic string transformations. While clean, the setting is far from real RL pipelines. There’s no guarantee that the same mechanism (composition of atomic skills) is what happens in open-ended reasoning tasks.

The paper defines a “new skill” as the ability to compose existing ones, but that might not satisfy the broader notion of novel reasoning or conceptual abstraction. In essence, this could still be pattern recombination rather than true skill learning. It is still possible that these compositionality is stored in the base model, and RL is just a better way to rediscover that.

**Questions:**

See weakness, thank you

---

> ### Author Response · Authors · 2025-11-21
> **Response to Reviewer xBeE**
>
> We thank you for the thoughtful and positive review. We are very glad you found our setup "impressively controlled," our evidence "clear," and our writing "strong”. We address the concerns one by one.
>
> ---
>
> ### On Weakness 1: Synthetic Setting vs. Real-World Pipelines
>
> This is a key point raised by multiple reviewers. As you note, our setting is "clean" but "far from real RL pipelines." We have a general response on this exact methodological choice (please see our General Response, above).
>
> In short: we fully agree this was a trade-off. We deliberately prioritized a clean, decontaminated, and controlled environment in this paper to ensure our claims about *skill acquisition* were causally sound and not an artifact of pre-training. We believe this foundational proof-of-concept is a necessary first step before applying this analysis to more complex, "real-world" (and harder-to-control) pipelines.
>
> ---
>
> ### On Weakness 2: Definition of "New Skill" and Latent Abilities
>
> We thank the reviewer for this insightful comment. Below, we will answer your two critical points: (1) whether "composition" constitutes a genuinely "new skill" versus a "meta-skill" or "conceptual abstraction," and (2) whether RL is merely rediscovering a latent capability, i.e., the composition.
>
> **1. On the Definition of "Skill" vs. "Composition"**
> We appreciate the nuance the reviewer highlights. Your understanding aligns with ours that new skills refer to a **"meta-skill"** or a form of **conceptual abstraction**, rather than a domain-specific skill (like calculation, coding, etc.).
>
> However, our terminology is explicitly grounded in **Anderson’s cognitive theory of skill acquisition** (Line 89 in our paper) [1]. Anderson posits that "knowledge compilation", i.e., the transition from declarative knowledge to fluid procedural skill, fundamentally relies on **composition**, "which collapses sequences of productions into single productions."
>
> In this view, the ability to recombine existing atomic operations into complex, stable workflows *is* the mechanism of skill learning. Our work demonstrates that RL is the driver of this "compilation" process. Even if one views this strictly as "pattern recombination," our results refute the prevailing view that RL is limited to simple re-ranking; we show that RL establishes the “glue” necessary to execute these combinations, which is important for many complex tasks. We believe it is important for the community to be aware of our conclusion.
>
> **2. On Latent Abilities and "Rediscovery"**
>
> We acknowledge that it is difficult to *definitively* prove that the "seed" of compositionality was not already "stored in the base model." As you suggested, the model can sample everything theoretically, and any claim of *absence* (i.e., "the model *cannot* do this") could be countered by saying one simply wasn't "lucky" enough to sample a correct compositional trajectory.
>
> Still, from Figures 2, 5, and 6, we tried our best to address the concern:
>
> 1. In our clean testbed, the base model (post-Stage 1) solves compositional problems with a really low accuracy (Figure 2).
> 2. Only **RL with compositional incentives** (our method) was able to reliably and systematically *activate* or *construct* this new skill, leading to strong generalization to deeper compositions (Figure 2, 5) and a fundamental change in reasoning behavior (Figure 6).
> 3. If the capability were merely latent and waiting to be "rediscovered," we would expect a non-trivial Pass@K score (indicating the solution exists in the probability landscape). The fact that extensive sampling fails to find correct highly-compositional trajectories (Figure 5) strongly suggests that this capability is *not* effectively present in the base model distribution. RL does not just "find" these paths; it constructs the necessary computational structure to generate them reliably.
>
> [1] Anderson, John R. "Acquisition of cognitive skill." *Psychological review* 89.4 (1982): 369.

---

### Official Review · Reviewer_oiQ7 · 2025-10-31

**Soundness:** 3
**Presentation:** 3
**Contribution:** 2
**Rating:** 6
**Confidence:** 4

**Summary:**

This paper argues that reinforcement learning (RL) can teach large language models genuinely new reasoning abilities by composing existing skills, not just reweighting old ones. Using a clean string-transformation testbed, the authors show that when models first master atomic skills and then receive RL on compositional tasks, they generalize to unseen and deeper compositions. The learned compositional skill even transfers to other domains like arithmetic reasoning, provided the model already has the basic atomic knowledge.

**Strengths:**

The paper offers an original and well-grounded perspective on the role of RL in LLMs, framing it as a process of compositional skill acquisition rather than simple reward reweighting. Its experimental design is exceptionally clean and controlled, allowing for causal interpretations. The writing is clear and logically structured, with a thoughtful connection to cognitive theories of human skill formation that strengthens its conceptual depth. The results are robust and systematically analyzed, making a compelling case that RL can produce genuinely new reasoning capabilities.

**Weaknesses:**

While the paper is conceptually strong, several aspects could be improved to strengthen its empirical and interpretive depth. First, the experimental validation is confined to synthetic string tasks and the Countdown domain which is clean but showing similar compositional learning in more natural reasoning settings would make the claims far more compelling.

**Questions:**

To what extent does model scale influence the emergence of compositionality? It would be helpful to know whether the same behaviors hold for smaller or larger models, as this would indicate whether compositional skill learning is a general property of RL or dependent on scale.

---

> ### Author Response · Authors · 2025-11-21
> **Response to Reviewer oiQ7**
>
> We thank the reviewer for the positive assessment and insightful summary. We address the concerns you raised in the review.
>
> ---
>
> ### Scope of Experiments & Model Scale
>
> Regarding the choice of a synthetic testbed, we refer the reviewer to our **General Response**, where we detail why a controlled, decontaminated environment is essential for making causal claims about learning versus memorization [1, 2].
>
> And as noted in the General Response, we have validated our findings on Qwen-4B with consistent results. Furthermore, we emphasize that Llama-3.1-8B was selected because previous work indicates it is among the most difficult open models to improve via RLVR [2,3]. Demonstrating compositional learning on this "hard case", where gains are typically harder to realize, suggests that our conclusions should naturally extend to models that are more amenable to RL.
>
> ---
>
> [1] Wu, Mingqi, et al. "Reasoning or memorization? unreliable results of reinforcement learning due to data contamination." *arXiv preprint arXiv:2507.10532* (2025).
>
> [2] Shao, Rulin, et al. "Spurious rewards: Rethinking training signals in rlvr." *arXiv preprint arXiv:2506.10947* (2025).
>
> [3] Gandhi, Kanishk, et al. "Cognitive behaviors that enable self-improving reasoners, or, four habits of highly effective stars." *arXiv preprint arXiv:2503.01307* (2025).

---

### Official Review · Reviewer_to4c · 2025-11-02

**Soundness:** 2
**Presentation:** 2
**Contribution:** 2
**Rating:** 2
**Confidence:** 4

**Summary:**

The paper proposes an evaluation of an LLM on compositional coding tasks, mainly to understand what extra value RL actually adds to compositional task performance.
Overall idea is interesting, but the clarity and the depth of analysis are somewhat limited.

**Strengths:**

1. The paper identifies and designs a coding-based compositional task setup that seems to probe model behaviour in a somewhat controlled way.

2. Experiments do show (at least in some parts) that RL helps improve compositional performance, esp. when moving from simpler to more complex tasks.

3. Discussion of prior work and related efforts is well-placed.

**Weaknesses:**

1. The novelty feels rather limited — much of the evaluation and task protocol seems adapted from existing works. The only clear “new” thing is using RL for level-1 and level-2 training, which looks like a small extension.

2. Only one model (LLaMA-3.1) is used, and with just 1-2 kinds of tasks (string manipulation and countdown types). That’s too narrow to make strong claims.

3. The plots mostly differ only by acc., avg@16, and pass@k numbers... also why choose k = 1000 ? feels arbitrary.

4. It’s also unclear why RL-based training reduces the basic atomic skill performance — this is quite puzzling and not explained. Figure 6 and its description comes across as this failure mode is "Acceptable".

5. No theoretical or mathematical grounding provided, which makes the claims feel a bit unsupported.

6. The paper doesn’t provide a limitations section.

**Questions:**

1. The writing quality can definitely be improved — there are several typos, some awkward phrases (“perform decently well” or “may remain limited new skills” etc.) that are confusing.

2. Would be nice if the focus didn’t stay only on compositional tasks… maybe discuss generalization to unseen atomic tasks too.

3. The connection to in-context or few-shot learning isn’t clear; authors do not mention how this fits into this line of work.

4. Why only LLaMA 3.1? adding more models could make the work more solid and convincing.

5. It might also help to check if a mix of RL and RFT training gives different outcomes — that could add an extra angle of analysis.

---

> ### Author Response · Authors · 2025-11-21
> **Response to Reviewer to4c (1/3)**
>
> We thank the reviewer for the detailed feedback. Below, we clarify several critical misunderstandings regarding our motivation, experimental design, and key findings.
>
> ---
>
> ### On Weakness 1: Novelty
>
> This paper aims to answer the heated debate on whether LLMs learn skills during RL. We conducted systematic experiments to rigorously disprove the claim. Note that the paper claiming RL does not incentivize reasoning achieved the only full score at NeurIPS 2025 and triggered an overwhelming discussion [4]. Therefore, we argue that our paper poses an in-time and very important contrast to a common belief in academia. Our novelty lies not only in the experimental framework but, more importantly, in the **novel insights and fundamental knowledge** we provide to the community:
>
> 1. **Challenging the Common Belief:** We challenge the prevailing pessimistic view that "RL does not teach new capabilities" [4]. Our work provides the rigorous empirical evidence needed to refute this belief, demonstrating that RL *can* strictly expand a model's reasoning horizon through composition.
> 2. **Quantifying Compositional Learning:** We provide new knowledge that **quantifies** the compositional learning behaviors of RL, showing exactly *how* and *when* new skills emerge (i.e., the necessity of atomic prerequisites).
> 3. **Fresh Insights on Post-Training:** We offer fresh insights into the design choices for post-training pipelines (specifically, the separation of atomic skill acquisition and compositional RL), providing a blueprint for achieving generalizable RL.
>
> Methodologically, these insights were made possible by our rigorous design of a controlled framework that strictly **controls for prior exposure** (confounding factors in pre-training). This allows us to isolate the "new skill" acquisition from mere retrieval, a distinction that standard benchmarks cannot make.
>
> ---
>
> ### On Weakness 2 & Question 4: Limited Models and Tasks
>
> We appreciate the suggestion to include more models and tasks. As detailed in our **General Response**, we chose this specific setup for the following reasons:
>
> 1. **Scientific Rigor:** To strictly control for **prior exposure** (a significant confounder in RL analysis [1, 2, 3]), we prioritized a controlled, synthetic environment over "real-world" tasks where pre-training data is unknown.
> 2. **Model Selection:** We selected Llama-3.1-8B not only because it is identified as a "cleaner" testbed [1, 2], but also because recent work [6] suggests Llama is known to be *less* responsive to RL than other families. Demonstrating skill acquisition on a model with weaker RL priors makes our conclusion more robust.
> 3. **Task Design:** Synthesizing tasks ensures the atomic skills are explicitly defined and the compositional logic is novel, preventing the model from simply retrieving memorized solutions.
>
> Regarding the model scale, as mentioned in our **General Response**, we have supplemented our paper with a new experiment on **Qwen-3-4B**. The results follow the exact same trend: the RL Level 1 model fails to generalize, while compositional RL unlocks performance on deeper levels. This confirms our conclusion is generalizable across different model families.
>
> ---
>
> ### On Weakness 3: Plots and Choice of *k*
>
> We clarify that the metrics and presentation of our results were targeted to answer distinct and critical research questions. While the axes may appear similar, the comparisons they visualize reveal fundamentally different phenomena:
>
> - **Figure 2 (Data Necessity):** Isolates the effect of training data. It indicates that RL on atomic tasks (Level 1) fails to teach composition, while RL on basic compositional tasks (Level 2) succeeds and generalizes.
> - **Figure 3 (Algorithm Necessity):** Isolates the learning method. It demonstrates that RL is the key ingredient; RFT fails to learn the compositional skill even when given the same data.
> - **Figure 4 (Transferability):** Tests cross-domain generalization. It shows the skill transfers to Countdown if and only if atomic skills are present (using Avg@32 to reduce variance).
> - **Figure 5 (Mechanism):** Directly refutes the "RL merely reranks" hypothesis. By visualizing Pass@k across difficulty levels, it shows the gap massively widens on hard problems (Levels 3+), proving the acquisition of a new skill rather than mere reranking.
>
> Regarding the choice of $k=1000$, we followed the **standard practice** in recent literature [4, 5] to ensure comparability. As stated in lines 419-421, this represents a "sufficiently large and practically meaningful budget" beyond which evaluation becomes impractical.

---

> > ### Author Response · Authors · 2025-11-21
> > **Response to Reviewer to4c (2/3)**
> >
> > ### On Weakness 4: Atomic Skill Performance
> >
> > We believe it’s the **misinterpretation of Figure 6** that leads to the ****claim of "RL-based training reduces the basic atomic skill performance".
> >
> > 1. **Atomic Skill is NOT Reduced:** As shown in **Figure 2 (Level 1 plot)**, our RL-trained models (e.g., `RL Level 1+2`, red line) maintain a high $\approx$80-90% accuracy on Level 1 (atomic) tasks. Atomic performance is preserved at a high level.
> > 2. **Figure 6 presents relative proportion, not absolute numbers:** Figure 6 shows the **proportion** *of failure modes of a model* on *difficult Level 3 compositional problems*. An "Atomic Error" is *only* counted if the model *first* succeeds at the compositional reasoning (i.e., it does *not* ignore composition or misinterpret nesting). Therefore, the “atomic error” of other models is not yet exposed and counted due to the lack of compositional capability. We have made it clearer in Figure 6’s caption.
> >
> > ---
> >
> > ### On Weakness 5: Theoretical Grounding
> >
> > We thank the reviewer for this point. We would like to clarify that instead of a mathematical derivation, our **theoretical grounding** is drawn from cognitive science, similar to seminal RL developments like *World Model* [7] or *Hindsight Experience Replay* [8]. We explicitly grounded our framework in the cognitive theory of human skill acquisition (Line 89), which posits that new skills are acquired by composing existing ones.
> >
> > Our contribution is the **empirical verification** of this cognitive hypothesis within the context of RL for LLMs. By isolating this mechanism in a controlled synthetic framework, our results provide concrete evidence that advances the community's understanding of how LLMs can evolve beyond simple reranking to true skill composition.
> >
> > ---
> >
> > ### On Weakness 6: Limitations Section
> >
> > Thank you for pointing that out. We have added a limitations section to the version, discussing the trade-off of using a synthetic setting for causal-discovery versus real-world (but contaminated) scenarios, as mentioned above.
> >
> > ---
> >
> > ### On Question 1: Writing
> >
> > Thank you for the careful reading. We have proofread the manuscript and corrected typos and awkward phrasing in the final version. We have uploaded the revised manuscript.
> >
> > ---
> >
> > ### On Question 2: Generalization to Unseen Atomic Tasks
> >
> > We actually have discussed this topic in **Section 4.3** and **Figure 4**. Our experiment is designed to test this.
> >
> > - The **`String-Base + RL L1+2`** model (green bar) shows that the compositional skill *does not* transfer to the Countdown task when the model *lacks the atomic skills* for Countdown.
> > - The **`Multi-Base + RL L1+2`** model (red bar) shows that when a model possesses the atomic skills for both tasks, RL training on Task A (string) *successfully transfers* the compositional meta-skill to Task B (Countdown).
> >
> > Besides, we also tried to examine the generalization to unseen atomic tasks on our string transformation task. However, we found it impossible in our setting, since our functions are given as meaningless function names like “func_1”. Therefore, there is no way for LLMs to figure out the function during RL and inference time to complete composition.

---

> > > ### Author Response · Authors · 2025-11-21
> > > **Response to Reviewer to4c (3/3)**
> > >
> > > ### On Question 3: Connection to In-Context Learning
> > >
> > > Thank you for the suggestion, but we don’t think our work focuses on or relates to in-context learning. We believe there may be a misunderstanding of our **Stage 1 training protocol**. As shown in Figure 7 and described in Section 3.2, this is a Rejection Fine-Tuning stage where the model is fine-tuned on its own correct reasoning trajectories to *internalize* atomic skills in its weights. We do *not* provide a few-shot demonstration in the prompt.
> > >
> > > Overall, we focus on RL and RFT, and the conclusion is orthogonal to ICL or few-shot learning. It is unclear how ICL or few-shot learning are immediately relevant in this context. We remain open to further discussion if the reviewer has specific insights on potential connections.
> > >
> > > ---
> > >
> > > ### On Question 5: Mix of RL and RFT Training
> > >
> > > We are not sure if we fully follow the suggestion, so we would be grateful if the reviewer could clarify the motivation and setup.
> > >
> > > Our paper focuses on three research questions (around Line 53): 1) Does RL teach new skills? 2) How can this learning be incentivized? and 3) Are these learned skills generalizable?
> > >
> > > We have shown that RFT can hardly contribute to the learning of compositional ability (Figure 3). We do not see the necessity for looking at the mix of RL and RFT training for this specific skill. Maybe RFT could be a promising approach for learning some new atomic skills with proper prompting, but that deviates from our current focus on *compositional* skill acquisition. We would be open to the input from the reviewer to supplement the paper.
> > >
> > > ---
> > >
> > > [1] Wu, Mingqi, et al. "Reasoning or memorization? unreliable results of reinforcement learning due to data contamination." *arXiv preprint arXiv:2507.10532* (2025).
> > >
> > > [2] Shao, Rulin, et al. "Spurious rewards: Rethinking training signals in rlvr." *arXiv preprint arXiv:2506.10947* (2025).
> > >
> > > [3] Wang, Yiping, et al. "Reinforcement learning for reasoning in large language models with one training example." *arXiv preprint arXiv:2504.20571* (2025).
> > >
> > > [4] Yue, Yang, et al. "Does reinforcement learning really incentivize reasoning capacity in llms beyond the base model?, 2025." *URL https://arxiv. org/abs/2504.13837*.
> > >
> > > [5] Wu, Fang, and Yejin Choi. "The invisible leash: Why rlvr may not escape its origin." *2nd AI for Math Workshop@ ICML 2025*. 2025.
> > >
> > > [6] Gandhi, Kanishk, et al. "Cognitive behaviors that enable self-improving reasoners, or, four habits of highly effective stars." *arXiv preprint arXiv:2503.01307* (2025).
> > >
> > > [7] Ha, David, and Jürgen Schmidhuber. "World Models." *CoRR* (2018).
> > >
> > > [8] Andrychowicz, Marcin, et al. "Hindsight experience replay." *Advances in neural information processing systems* 30 (2017).

---

### Author Response · Authors · 2025-11-21
**General Response**

## General Response 1: Justification for Synthetic Setting

We thank the reviewers for raising this important point regarding the scope of our experiments. We agree that demonstrating these findings in more complex, "natural" reasoning settings is a valuable and important direction for future work. However, we argue that using a synthetic task is arguably the only feasible choice for the goal of **scientific rigor and causal isolation** regarding the research question in this paper.

The central question of our paper, i.e., whether RL *genuinely teaches* new skills or just *activates existing ones,* requires an experimental setup that strictly ablates **prior exposure**. On most of the tasks, it is nearly impossible to determine if a model is learning a genuinely new compositional skill or merely activating a latent capability acquired during pre-training. As noted by previous work, many base models have likely been exposed to vast amounts of similar reasoning patterns or even the test data, making them "confounded" testbeds for studying the *origin* of a skill [1, 2].

Our *entire* experimental framework was designed with **capability isolation as a core principle**. By selecting a model identified as having a "cleaner" baseline for these specific types of reasoning (Llama-3.1-8B) and designing novel, synthetic tasks from scratch, we create a controlled environment where the **atomic skills** are explicitly defined and the **compositional logic** is novel to the model. This almost ensures that the compositional skill we observe is *acquired* during our Stage 2 training, rather than being "rediscovered" from the pre-training distribution.

---

## General Response 2: Model Scale & Generalization

To further address the concern that our findings might be specific to the model choice or scale, during the rebuttal period, we replicated our main experiment on **Qwen-3-4B**, a model with a different architecture and scale. We kindly refer the reviewers to the figure in our anonymized repository [`https://anonymous.4open.science/r/rl-composition-rebuttal-DABE/qwen3_val.pdf`](https://anonymous.4open.science/r/rl-composition-rebuttal-DABE/qwen3_val.pdf). **The results follow the exact same trend:** RL on atomic tasks fails to generalize to high-level tasks, while RL on basic Level-2 compositional tasks unlocks generalization to deeper levels. This consistency across model families and scales reinforces that our synthetic testbed is capturing a fundamental mechanism of RL-based skill acquisition, rather than an artifact of a specific model.

We believe that providing this clear, causal proof-of-concept is a necessary and foundational contribution. We have established that the compositional mechanism *is* possible and LLMs learn new skills in RL through this mechanism. We hope this provides a solid foundation for the community to investigate it in more complex settings.

[1] Wu, Mingqi, et al. "Reasoning or memorization? unreliable results of reinforcement learning due to data contamination." *arXiv preprint arXiv:2507.10532* (2025).

[2] Shao, Rulin, et al. "Spurious rewards: Rethinking training signals in rlvr." *arXiv preprint arXiv:2506.10947* (2025).

---

### Meta-Review · Area_Chair_WsCL · 2026-01-08

**Summary:**

The paper investigates whether RL can teach LLMs genuinely new skills, operationalized as learning to solve unseen compositions of already-mastered atomic string transformation functions. Results suggest RL on compositional data (after atomic skill acquisition) yields generalization to deeper compositions. Reviewers agree the question is timely and the controlled setup is a strength.

Concerns remain about external validity beyond synthetic tasks, whether “composition” here constitutes a sufficiently strong notion of “new skill” versus recombination/rediscovery, limited breadth of evidence (tasks/models), and gaps in methodological reporting/justification. The rebuttal clarifies some points and adds an additional model replication.

Despite these concerns, the work is overall well-executed and an important contribution to the literature.

**Reviewer Concerns:**

Reviewer to4c:
Addressed (partially): clarified novelty and contributions; clarified content of Figure 6; added limitations + proofreading; added an extra model replication.
Outstanding: Empirical scope still narrow (limited model/task diversity). Theoretical/mechanistic grounding remains thin.

Reviewer oiQ7
Addressed: Extra model replication and discussion of why synthetic setting is used for causal isolation.
Outstanding: Still lacks evidence in more naturalistic reasoning settings; scale dependence not systematically addressed.

Reviewer xBeE
Addressed: Rebuttal engages “latent rediscovery vs new skill” critique and defends synthetic trade-off.
Outstanding: external validity to real RL-for-reasoning pipelines remains unproven.

Reviewer UU3i
Addressed: Stage 1 is sufficient but not proven necessary; added discussion. Provided responses on entropy/KL settings and fairness framing for RL vs RFT; reviewer indicates score unchanged.
Outstanding: Key alternative (RL learning atomic+compositional jointly) not empirically tested.

**Reviewer Scores:**

to4c: 2->3 or 4
oiQ7: likely unchanged (6)
xBeE: likely unchanged (6)
UU3i: unchanged (6), explicitly stated.

---

### Decision · Program_Chairs · 2026-01-26

Accept (Poster)